# Molecular Networking-Guided Annotation of Flavonoid Glycosides from *Quercus mongolica* Bee Pollen

**DOI:** 10.3390/ijms26167930

**Published:** 2025-08-17

**Authors:** Yerim Joo, Eunbeen Shin, Hyunwoo Kim, Mi Kyeong Lee, Seon Beom Kim

**Affiliations:** 1Department of Food Science and Technology, Pusan National University, Miryang 50463, Republic of Korea; fapnu.yrj@gmail.com (Y.J.); fapnu.ebs@gmail.com (E.S.); 2Institute for Future Earth, Pusan National University, Busan 46421, Republic of Korea; 3College of Pharmacy and Integrated Research Institute for Drug Development, Dongguk University-Seoul, Goyang 10326, Republic of Korea; hwkim8906@dongguk.edu; 4College of Pharmacy, Chungbuk National University, Cheongju 28160, Republic of Korea; 5Food Tech Innovation Center, Life and Industry Convergence Research Institute, Pusan National University, Miryang 50463, Republic of Korea

**Keywords:** *Quercus mongolica*, pollen, molecular networking, LC–MS/MS, flavonoids

## Abstract

Bee pollen is a primary and secondary metabolite-rich natural product collected by pollinators such as honeybees. Polyphenols, particularly flavonoids, are well known for their potent antioxidant activities. Numerous phytochemical and biological studies have focused on *Quercus mongolica*, a member of the Fagaceae family. However, research focusing specifically on pollen is limited. Moreover, bee pollen chemical composition varies significantly depending on its geographical origin and cultivation conditions. In this study, the flavonoid glycosides of *Q. mongolica* pollen were profiled using LC–MS/MS-based molecular networking, which revealed that the largest molecular cluster corresponded to flavonoid glycosides. A total of 69 flavonoid glycosides, primarily comprising 2 kaempferol derivatives, 14 quercetin derivatives, and 46 isorhamnetin derivatives, were annotated based on MS/MS fragmentation patterns, spectral library matches in GNPS (Global Natural Products Social Molecular Networking), and comparison with previously reported data. Two primary compounds, isorhamnetin 3-*O*-*β*-_D_-xylopyranosyl (1→6)-*β*-_D_-glucopyranoside and isorhamnetin-3-*O*-neohesperidoside, were identified by comparison with reference standards. This study offers foundational insights into the flavonoid diversity of *Q. mongolica* pollen, contributing to a broad understanding of its secondary metabolite profile.

## 1. Introduction

Floral pollen is a male gametophyte produced in the anthers of flowers and mainly transferred by pollinators, such as honeybees, during pollination. Honeybees collect floral pollen and mix it with their secretions and nectar, forming bee pollen [1,2,3,4,5]. Bee secretions contain various enzymes such as amylase, invertase, and glucosidase, which assist in binding and compacting the pollen [4]. During this transformation, bee pollen may exhibit morphological and chemical characteristics distinct from the original floral pollen [6]. Bee pollen is used as a raw material for pharmaceuticals and cosmetics, and mainly consumed as a health supplement [1,7,8].

Previous studies have identified essential oils, phenolic compounds, polyamines, and their metabolites in bee pollen [7,9,10]. Among these, polyphenols, particularly flavonoids, contribute significantly to the potent antioxidant activity of bee pollen [1,8]. Flavonoids have attracted considerable attention for their diverse bioactivities, including antioxidant, anti-inflammatory, antibacterial, and anticancer effects [11,12]. Also, our previous research has demonstrated that flavonoids derived from *Quercus mongolica* pollen exhibit significant antioxidant and tyrosinase-inhibitory activities, underscoring their potential application in health-promoting products [13]. Given this context, understanding the flavonoid composition of bee pollen can aid in the identification of novel bioactive compounds beneficial for pharmaceutical and nutraceutical applications.

Flavonoids are recognized as the major chemical constituents of plant pollen, predominantly occurring as flavonoid glycosides [7,8,14,15]. These flavonoid glycosides exhibit remarkable structural complexity and diversity, attributed to differences in glycosylation sites, linkage types, and sugar moieties. Glycosylation improves the chemical stability of flavonoids and increases the aqueous solubility of the aglycone backbone, thereby improving their bioavailability [15,16,17,18,19]. In this study, we primarily focused on bee pollen-derived flavonoid glycosides.

*Quercus mongolica*, a deciduous hardwood tree in the Fagaceae family, occurs broadly throughout Republic of Korea and accounts for approximately 3.7% of the country’s forested area [1,20,21,22,23]. In a previous study, 18 flavonoids were isolated and identified from *Q. mongolica* pollen, 11 of which were flavonoid glycosides. Quercetin, kaempferol, and isorhamnetin were predominantly present as their glycosides in *Q. mongolica* pollen, and these compounds exhibited potent antioxidant activity [13]. Numerous reports have focused on flavonoids and their glycosides in bee pollen. However, various flavonoid glycosides that protect pollen from external environmental factors have not been identified yet. The discovery of these compounds is significant, but research on flavonoid glycosides in bee pollen remains underdeveloped.

Molecular networking using the Global Natural Products Social Molecular Networking (GNPS) platform has become a powerful approach for untargeted metabolomics in natural product studies. This technique clusters compounds based on similar MS/MS fragmentation patterns, enabling visualization of their molecular relationships as networks, and facilitating efficient metabolite annotation through spectral libraries [24,25,26].

Research on the flavonoid chemical profiling of *Q. mongolica* pollen is limited. Moreover, the chemical composition of bee pollen can vary depending on geographic location and cultivation practices [7,8,27,28]. Therefore, this study aimed to conduct a preliminary chemical profiling of *Q. mongolica* pollen using molecular networking. This approach provides valuable insights into the flavonoid glycoside diversity and potential bioactivity, offering a valuable foundation for subsequent research and applications.

## 2. Results

### 2.1. Flavonoid Glycoside Profiling of Q. mongolica Pollen Using Molecular Networking Based on UPLC–QTOF–MS/MS Analysis

GNPS based on UPLC–QTOF–MS/MS. The MS/MS data were acquired in negative ion mode, which provides high selectivity and sensitivity for the LC–MS analysis of flavonoids in plant resources [29]. Figure 1 shows the base peak ion chromatogram (BPI) of *Q. mongolica* pollen. According to LC–MS/MS analysis, peaks eluting between 3.5 and 6.0 min were identified as flavonoid glycosides, whereas those between 6.0 and 7.8 min were attributed to polyamine compounds. Three major flavonoid glycosides were shown on the base peak ion chromatogram (BPI) of *Q. mongolica* pollen, in which three primary flavonoid glycosides were prominently observed (A1–A3).

In this study, *Q. mongolica* pollen metabolites were profiled using molecular networking, a technique that facilitates the discovery of untargeted metabolites based on LC–MS/MS data. While the BPI data contained a vast amount of information that is difficult to interpret manually, molecular networking provided an intuitive visualization, enabling the identification of target compounds and their related derivative compounds. In the case of flavonoid glycosides, the MS/MS fragmentation patterns revealed distinct signals corresponding to the aglycone part (flavonoid) and the sugar moiety. By analyzing these patterns, it was confirmed that the structures could be categorized into aglycone backbones and sugar components (monosaccharides and disaccharides). Furthermore, the relative intensity of ionized peaks was used to predict the substitution positions of sugar residues, thereby assisting in the structural identification of the compounds. To annotate the flavonoid glycosides, molecular networking analysis was performed based on the UPLC–QTOF–MS/MS data using the GNPS platform. Two major clusters were observed in the MS/MS spectral network of *Q. mongolica* pollen (Figure 2). Based on the spectral library matches in the GNPS, Cluster A was identified as a flavonoid glycoside, whereas Cluster B was classified as a polyamine. In total, 69 flavonoid glycosides were grouped into Cluster A, with a predominance of flavonoid *O*-glycosides. Among the annotated compounds, kaempferol, quercetin, and isorhamnetin were identified as major aglycone backbones. Figure 3 shows that two primary compounds, isorhamnetin 3-*O*-*β*-_D_-xylopyranosyl (1→6)-*O*-*β*-_D_-glucopyranoside and isorhamnetin 3-*O*-neohesperidoside, were confirmed using ^1^H and ^13^C NMR and LC–MS spectroscopic data (Appendix A). The chemical composition of bee pollen varies depending on geographic origin, collection period, and cultivation conditions, factors which are also known to influence its biological activities. Therefore, to utilize the natural resource of bee pollen, standardization studies are required. Two primary compounds, isorhamnetin 3-*O*-*β*-_D_-xylopyranosyl (1→6)-*O*-*β*-_D_-glucopyranoside and isorhamnetin 3-*O*-neohesperidoside, can be used as standard compounds for future standardization research. These compounds were selected because they are among the major constituents of *Q. mongolica* pollen, and their chromatographic signals were well separated from other metabolite peaks, making them suitable for accurate qualitative and quantitative analysis.

Research on Flavonoid glycosides exhibits characteristic fragmentation patterns involving both aglycone and sugar moieties. The sugar moieties were primarily hexose (−162 Da), pentose (−132 Da), and deoxyhexose (−146 Da). In some cases, a neutral loss of hexose along with a water molecule (−180 Da) was observed. Additionally, some glycosides were acetylated, exhibiting fragment losses corresponding to an acetyl group (−42 Da), acetic acid (−60 Da), or acetylated hexose (−204 Da) [24,30]. All compounds were tentatively annotated based on spectral matching with the GNPS library, comparison with reference compounds, and results of previous studies [30,31].

To determine the glycosylation sites of flavonoid glycosides, we compared the relative intensities of deprotonated and radical aglycone ions. In flavonoid *O*-glycosides, these fragment ions are typically generated during glycoside fragmentation under negative ion mode [31]. However, the abundance ratio between these ions can vary depending on the substitution pattern on the flavonoid backbone, glycosylation site, and sugar moiety type. Glycosylation at the 3-OH position tends to increase the abundance of radical aglycone ions, whereas glycosylation at the 7-OH position increases the abundance of deprotonated aglycone ions [31,32,33]. Based on these characteristic fragmentation patterns, the glycosylation positions of the annotated flavonoid glycosides were tentatively identified.

### 2.2. Identification of Kaempferol Derivatives

The annotated compounds are listed in Table 1. Compounds **1** and **2** were identified as kaempferol glucosides based on their characteristic product ions at *m*/*z* 285, 255, and 227 [29,32].

Compound **1** (RT 6.93 min) exhibited a deprotonated molecular ion ([M−H]^−^) at *m*/*z* 489.104. In the MS/MS spectrum, a fragment ion at *m*/*z* 285 indicated the neutral loss of acetylhexoside moiety (−204 Da) [34]. The appearance of a fragment ion at *m*/*z* 429 suggested the loss of an acetyl group (−60 Da). The glycosylation site was tentatively assigned to the 3-OH position based on the higher intensity of [M−H−204]^•−^ than that of [M−H−204]^−^. Consequently, compound **1** was putatively characterized as kaempferol 3-*O*-acetylglucoside [24].

Compound **2** (RT 4.69 min) presented a [M−H]^−^ at *m*/*z* 593.151 and generated a fragment ion at *m*/*z* 285, indicating the loss of deoxyhexose and pentose moieties (−308 Da). The 3-OH position was proposed as the glycosylation site, inferred from the relatively high intensity of the [M−H−308]^•−^ ion. Accordingly, compound **2** was putatively annotated as kaempferol 3-*O*-rutinoside based on GNPS library matches and comparison with the results of previous studies [24,35].

**Table 1 ijms-26-07930-t001:** Putative identification of flavonoid glycosides in the flavonoid glycoside cluster of *Quercus mongolica* pollen extracts analyzed by UPLC–QTOF–MS in negative ion mode.

No.	RT (min)	[M−H]^−^ (*m*/*z*)	Molecular Formula (Error in ppm)	Tentatively Identification	MS^2^ (*m*/*z*)	Ref
Kaempferol derivatives						
1	6.92	489.104	C_23_H_22_O_12_ (−0.1)	Kaempferol-3-*O*-acetylglucoside	429, 309, 285, **284**, **255**, **227**	[24]
2	4.69	593.151	C_27_H_30_O_15_ (−0.2)	Kaempferol-3-*O*-rutinoside ^a^	285, **284**, **255**, **227**	[24,35]
Quercetin derivatives						
3	4.95	463.088	C_21_H_20_O_12_ (0.0)	Quercetin-3-*O*-glucoside ^a^	**301**, **300**, **271**, 255, 243	[34,35]
4	6.03	505.098	C_23_H_22_O_13_ (−0.3)	Quercetin-3-*O*-acetylglucoside ^a^	**301**, **300**, **271**, 255, 243	[24]
5	6.27	573.086	C_33_H_18_O_10_ (4.9)	Quercetin derivatives	505, **301**, **300**, 271, 255	
6	4.17	595.132	C_26_H_28_O_16_ (1.2)	Quercetin-3-*O*-sambubioside ^a,b^	463, 445, **301**, **300**, **271**, 255	[36]
7	4.13	609.147	C_27_H_30_O_16_ (0.8)	Quercetin-3-*O*-rutinoside ^a^	463, 445, 301, **300**, **271**, **255**	[37]
8	3.70	625.142	C_27_H_30_O_17_ (1.1)	Quercetin-3-*O*-sophoroside ^a^	463, 445, 301, **300**, **271**, **255**	[36,37]
9	4.17	663.117	C_29_H_28_O_18_ (−4.1)	Quercetin derivatives	**595**, 323, **301**, **300**	
10	4.19	679.091	C_32_H_24_O_17_ (−5.1)	Quercetin derivatives	**611**, **595**, 301, **300**, 271	
11	3.72	693.128	C_37_H_26_O_14_ (4.2)	Quercetin derivatives	647, **625**, 323, **301**, **300**	
12	3.72	709.100	C_33_H_26_O_18_ (−6.3)	Quercetin derivatives	**625**, 399, 384, **301**, **300**	
13	4.27	725.193	C_29_H_26_O_22_ (5.9)	Quercetin derivatives	**679**, **633**, **384**, 301, 300, 284	
14	4.15	747.072	C_24_H_28_O_27_ (−3.4)	Quercetin derivatives	**701**, **655**, **595**, 523, 301, 300	
15	3.74	761.114	C_33_H_30_O_21_ (−9.3)	Quercetin derivatives	715, 647, **625**, **323**, **301**, 300	
16	3.72	777.082	C_32_H_26_O_23_ (3.6)	Quercetin derivatives	731, **685**, **625**, 523, 301, **300**	
Isorhamnetin derivatives						
17	6.46	461.109	C_22_H_22_O_11_ (−0.3)	Isorhamnetin-3-*O*-rhamnoside	315, **314**, 300, 299, 285, **271**, 257, **243**	[24]
18	5.79	477.104	C_22_H_22_O_12_ (1.2)	Isorhamnetin-3-*O*-glucoside isomer ^a^	315, **314**, **299**, 285, **271**, 257, 243	[38]
19	5.47	477.104	C_22_H_22_O_12_ (1.4)	Isorhamnetin-3-*O*-glucoside isomer ^a^	315, **314**, 300, **299**, **271**, 255	[38]
20	6.53	519.114	C_24_H_24_O_13_ (−0.2)	Isorhamnetin-3-*O*-acetylglucoside isomer	315, **314**, 299, **285**, **271**, 257, 243	
21	6.79	519.114	C_24_H_24_O_13_ (0)	Isorhamnetin-3-*O*-acetylglucoside isomer	315, **314**, 299, **285**, **271**, 257, 243	
22	7.11	519.114	C_24_H_24_O_13_ (0.7)	Isorhamnetin-3-*O*-acetylglucoside isomer	459, **315**, **314**, 300, 299, **285**, 271	
23	5.45	545.091	C_25_H_22_O_14_ (−3.3)	Isorhamnetin derivatives	**477**, 315, **314**, 300, **299**, 271	
24	5.77	545.091	C_32_H_18_O_9_ (5.3)	Isorhamnetin derivatives	**477**, **315**, **314**, 299, 285, 271,	
25	5.77	591.093	C_33_H_20_O_11_ (−0.3)	Isorhamnetin derivatives	**477**, **383**, 315, **314**, 285, 271	
26	6.58	607.167	C_28_H_32_O_15_ (0.8)	Isorhamnetin derivatives	315, **314**, **299**, 285, **271**	
27	5.79	607.061	C_25_H_20_O_18_ (5.8)	Isorhamnetin derivatives	**399**, **383**, 371, **315**, 314, 299	
28	5.47	609.148	C_27_H_30_O_16_ (1.5)	Isorhamnetin-3-*O*-*β*-_D_-xylopyranosyl(1→6)-*β*-_D_-glucopyranoside ^a,b^	**315**, **314**, **300**, 299, 271	
29	7.26	623.140	C_31_H_28_O_14_ (0.1)	Isorhamnetin derivatives	477, **315**, **314**, 300, **299**, 271	
30	4.75	623.162	C_28_H_32_O_16_ (1.2)	Isorhamnetin-3-*O*-neohesperidoside ^a,b^	**315**, **314**, 300, **299**, 285, 271	[24,38]
31	7.09	633.104	C_35_H_22_O_12_ (0.4)	Isorhamnetin derivatives	**519**, **383**, 315, **314**, 300, 299, 112	
32	4.08	639.158	C_28_H_32_O_17_ (−0.6)	Isorhamnetin-3-*O*-sophoroside ^a^	459, 315, **314**, **300**, **299**, 271	[39]
33	7.02	651.157	C_29_H_32_O_17_ (0.5)	Isorhamnetin derivatives	**591**, **315**, **314**, 299, 285, 243	
34	6.60	651.158	C_29_H_32_O_17_ (0.5)	Isorhamnetin derivatives	**519**, **383**, 315, **314**, 300, 299	
35	7.13	655.088	C_30_H_24_O_17_ (−8.4)	Isorhamnetin derivatives	**519**, 459, 315, **314**, 300, 299	
36	4.73	669.166	C_29_H_34_O_18_ (−1.1)	Isorhamnetin derivatives	**623**, **315**, **314**, 299	
37	4.12	675.132	C_34_H_28_O_15_ (−5.2)	Isorhamnetin derivatives	**639**, 321, 315, **314**, 300, **299**	
38	5.47	677.133	C_30_H_30_O_18_ (−5.2)	Isorhamnetin derivatives	**609**, **315**, **314**, 300, 299	
39	4.15	685.099	C_29_H_34_O_19_ (−1.0)	Isorhamnetin derivatives	**639**, 315, **314**, 300, **299**	
40	4.08	685.162	C_29_H_34_O_19_ (−1.2)	Isorhamnetin derivatives	**639**, 315, **314**, 300, **299**	
41	4.73	691.149	C_31_H_32_O_18_ (−4.2)	Isorhamnetin derivatives	**623**, 337, **315**, **314**	
42	5.47	693.105	C_33_H_26_O_17_ (−6.5)	Isorhamnetin derivatives	647, **609**, 413, **315**, **314**	
43	7.07	697.169	C_23_H_26_O_17_ (−0.5)	Isorhamnetin derivatives	**651**, 605, 591, **315**, **314**, 300, 299	
44	7.15	697.198	C_31_H_38_O_18_ (−3.2)	Isorhamnetin derivatives	**651**, 605, 591, **315**, **314**, 299	
45	4.10	707.143	C_38_H_28_O_17_ (−3.2)	Isorhamnetin derivatives	**639**, 609, **315**, **314**, 300, 299	
46	7.00	719.144	C_32_H_32_O_19_ (−3.6)	Isorhamnetin derivatives	**651**, **591**, 315, **314**	
47	5.47	723.138	C_38_H_28_O_15_ (3.9)	Isorhamnetin derivatives	**631**, **609**, **315**, 314	
48	4.73	737.154	C_39_H_30_O_15_ (3.8)	Isorhamnetin derivatives	691, **645**, **623**, 315, **314**	
49	4.92	739.104	C_23_H_32_O_27_ (−3.5)	Isorhamnetin derivatives	**693**, **609**, **399**, 399, 315, 314	
50	5.47	739.104	C_23_H_32_O_27_ (−2.5)	Isorhamnetin derivatives	**693**, **647**, 609, 479, **413**, 315	
51	5.47	745.120	C_33_H_30_O_20_ (−8.1)	Isorhamnetin derivatives	**653**, **609**, **315**, 314	
52	4.8	753.119	C_24_H_34_O_27_ (−2.9)	Isorhamnetin derivatives	**707**, 661, **623**, **399**, 315, 314	
53	4.1	753.148	C_39_H_30_O_16_ (2.6)	Isorhamnetin derivatives	707, **639**, **315**, **314**, 300, 299	
54	5.49	761.089	C_25_H_30_O_27_ (−2.1)	Isorhamnetin derivatives	669, 663, **609**, **435**, **315**, 314	
55	7.05	765.149	C_33_H_34_O_21_ (−4.2)	Isorhamnetin derivatives	701, 673, **651**, **315**, **314**, 112	
56	4.08	775.130	C_34_H_32_O_21_ (−8.4)	Isorhamnetin derivatives	**639**, **609**, 315, **314**, 300, 299	
57	7.00	781.116	C_25_H_34_O_28_ (−1.0)	Isorhamnetin derivatives	**689**, 651, **629**, **413**, 315, 314	
58	4.12	791.099	C_26_H_32_O_28_ (−2.4)	Isorhamnetin derivatives	699, **639**, **609**, 315, 314, 301, **300**, 299	
59	5.49	807.090	C_33_H_28_O_24_ (−0.3)	Isorhamnetin derivatives	**715**, **669**, 609, **435**, 315, 314	
60	4.75	811.106	C_29_H_32_O_27_ (−0.1)	Isorhamnetin derivatives	765, **623**, 456, **315**, **314**	
61	4.34	823.191	C_36_H_40_O_22_ (−3.7)	Isorhamnetin derivatives	**755**, **315**, **314**	
62	4.08	837.101	C_34_H_30_O_25_ (0.9)	Isorhamnetin derivatives	**723**, **699**, **639**, 399, 315, 314	
Quercetagetin-dimethyl derivatives						
63	5.64	507.114	C_23_H_24_O_13_ (0.2)	Quercetagetin-dimethyl 3-*O*-hexoside	**345**, 344, 330, **329**, 314, **301**, 286	[40]
64	5.65	637.071	C_26_H_22_O_19_ (5.1)	Quercetagetin-dimethyl derivatives	**429**, **413**, **386**, 345, 329, 300	
65	4.50	653.172	C_29_H_34_O_17_ (−0.1)	Quercetagetin-dimethyl derivatives	345, **344**, 330, **329**, **301**	
Others						
66	5.62	575.101	C_26_H_24_O_15_ (−4.8)	unknown	**507**, 492, **344**, **329**, 301	
67	3.34	621.146	C_28_H_30_O_16_ (−0.2)	unknown	327, **326**, 312, **311**, 284, 283	
68	4.15	725.089	C_29_H_26_O_22_ (5.9)	unknown	**679**, **633**, 595, 399, **384**, 301	
69	3.74	755.098	C_30_H_28_O_23_ (4.7)	unknown	**709**, **663**, 625, 399, **384**, 301	

^a^ Compounds were tentatively identified based on GNPS spectral library matches; ^b^ Identified by comparison with reference standards. The most abundant fragment ions are shown in **bold**.

### 2.3. Identification of Quercetin Derivatives

Characteristic fragment ions of quercetin were observed at *m*/*z* 301, 271, and 255 [29,32] Compound **3** (RT 4.95 min) exhibited a [M−H]^−^ at *m*/*z* 463.088 and produced a fragment ion at *m*/*z* 301, indicating the loss of a hexose unit (−162 Da). The predominance of the [M−H−162]^•−^ ion suggests glycosylation of 3-OH. Consequently, compound **3** was putatively characterized as quercetin 3-*O*-glucoside, in agreement with the GNPS library matches and results of previous reports [34,35]. Compound **4** (RT 6.03 min) showed a [M−H]^−^ at *m*/*z* 505.098 and generated a fragment ion at *m*/*z* 301, indicating the loss of an acetylhexoside moiety (−204 Da). The 3-OH position was also proposed as a glycosylation site based on the relatively high intensity of the [M−H−204]^•−^ ion. Consequently, compound **4** was putatively characterized as quercetin 3-*O*-acetylglucoside, supported by the GNPS library matches and results of previous studies [24]. Compound **6** (RT 4.17 min), identified as primary compound A1, exhibited a [M−H]^−^ at *m*/*z* 595.131. As shown in the MS/MS spectrum in Appendix A, a fragment ion at *m*/*z* 301 indicated the loss of hexose and pentose moieties (−294 Da), and an additional fragment at *m*/*z* 463 corresponded to the loss of a pentose unit (−132 Da). The predominance of the [M−H−294]^•−^ ion suggested glycosylation at the 3-OH position. Consequently, compound **6** was putatively annotated as quercetin 3-*O*-sambubioside [36]. Compound **7** (RT 4.13 min) showed a [M−H]^−^ at *m*/*z* 609.147 and produced a fragment ion at *m*/*z* 301, consistent with the loss of deoxyhexose and pentose moieties (−308 Da). Additional fragment ions at *m*/*z* 463 and 445 were observed, indicating the loss of a deoxyhexose moiety (−146 Da) and a hexose moiety (−162 Da), respectively. The 3-OH position was tentatively suggested to be the glycosylation site, supported by the predominance of [M−H−308]^•−^. Consequently, compound **7** was putatively annotated as quercetin 3-*O*-rutinoside based on GNPS library matches and comparison with the results of previous studies [38]. Compound **8** (RT 3.70 min) exhibited a [M−H]^−^ at *m*/*z* 625.142 and generated a fragment ion at *m*/*z* 301, indicating the loss of dihexose moieties (−324 Da). A fragment ion at *m*/*z* 465, indicating the loss of a single hexose (−162 Da). The glycosylation site was tentatively determined to be at the 3-OH position, based on the higher intensity of the [M−H−324]^•−^ ion. Consequently, compound **8** was putatively characterized as quercetin 3-*O*-sophoroside, supported by GNPS library matches and results of previous reports [36,37]. Compounds **9**–**16** were provided as quercetin glycoside derivatives, and it is anticipated that these compounds can be identified through isolation and purification based on the preliminary information presented.

### 2.4. Identification of Isorhamnetin Derivatives

The characteristic fragment ions of isorhamnetin were observed at *m*/*z* 300, 271, 255, and 227 [32] Compound **17** (RT 6.46 min) exhibited a [M−H]^−^ at *m*/*z* 461.109 and generated a fragment ion at *m*/*z* 315, indicating the loss of deoxyhexose (−146 Da). The glycosylation site was tentatively determined to be at the 3-OH position based on the higher intensity of the [M−H−146]^•−^ ion. Consequently, compound **17** was putatively characterized as isorhamnetin 3-*O*-rhamnoside based on comparison of the obtained MS/MS spectra with those reported in previous studies [24]. Compounds **18** (RT 5.79 min) and **19** (RT 5.47 min) exhibited identical [M−H]^−^ ions at *m*/*z* 477.104 and generated a fragment ion at *m*/*z* 315, indicating the loss of hexose (−162 Da). The glycosylation site was tentatively determined to be at the 3-OH position, based on the higher intensity of the [M−H−162]^•−^ ion. Accordingly, compounds **18** and **19** were putatively characterized as isorhamnetin 3-*O*-glucoside isomer supported by the GNPS library matches and previous reports [38]. Compounds **20** (RT 6.53 min), **21** (RT 6.79 min), and **22** (RT 7.11 min) displayed a [M−H]^−^ at *m*/*z* 519.114 and generated a fragment ion at *m*/*z* 315, indicating the loss of acetyl hexose [M−H−204]^−^. A fragment ion at *m*/*z* 459 corresponded to the loss of an acetyl group (−60 Da). The glycosylation site was tentatively determined to be at the 3-OH position, based on the higher intensity of the [M−H−204]^•−^ ion. Accordingly, compounds **20**, **21**, and **22** were putatively characterized as isorhamnetin 3-*O*-acetyl glucoside isomers. Compound **28** (RT 5.47 min), identified as primary compound A3, presented a [M−H]^−^ at *m*/*z* 609.148, and produced a fragment ion at *m*/*z* 315, indicating the loss of hexose and pentose moieties (−294 Da). As previously described, the tentative assignment of glycosylation positions can be inferred from a relative intensity comparison between the radical aglycone ions and regular deprotonated aglycone ions. However, these intensities may vary depending on the flavonoid backbone, glycosylation position, and the type of sugar moiety. For compound **28**, although the intensity of [M−H−294]^−^ was higher than that of [M−H−294]^•−^, it was ultimately confirmed as a 3-*O*-glycoside by NMR spectroscopy. This compound was confirmed by comparing its 1D (^1^H and ^13^C) and 2D (COSY, HSQC, HMBC) NMR spectrum with that of a reference standard (Appendix A). Accordingly, compound **28** was identified as isorhamnetin 3-*O*-*β*-_D_-xylopyranosyl(1→6)-*β*-_D_-glucopyranoside. Compound **30** (RT 4.75 min), identified as the primary compound A2, exhibited a [M−H]^−^ at *m*/*z* 623.162 and produced a fragment ion at *m*/*z* 315, indicating the loss of deoxyhexose and hexose moieties (−308 Da). Based on GNPS spectral library matching, this compound was initially annotated as isorhamnetin 3-*O*-robinoside. However, a comparison of the 1D (^1^H and ^13^C) and 2D (COSY, HSQC, HMBC) NMR spectra with those of a reference compound confirmed that the compound was isorhamnetin 3-*O*-neohesperidoside (Appendix A). These findings suggest that although GNPS spectral library matching is a useful tool for putative annotation, definitive structural elucidation requires complementary analytical techniques with both NMR and LC–MS/MS. Accordingly, **30** was identified as isorhamnetin 3-*O*-neohesperidoside [24,38]. Compound **32** (RT 4.08 min) presented a [M−H]^−^ at *m*/*z* 639.158 and generated a fragment ion at *m*/*z* 315, indicating the loss of dihexose moieties (−324 Da). A fragment ion at *m*/*z* 477, indicating the loss of a single hexose (−162 Da). The glycosylation site was tentatively determined to be at the 3-OH position, based on the higher intensity of the [M−H−324]^•−^ ion. Consequently, compound **32** was putatively characterized as isorhamnetin 3-*O*-sophoroside, as supported by the GNPS library matches and previous reports [39]. Compounds **23**–**27**, **29**, **31**, and **33**–**62** were provided as isorhamnetin glycoside derivatives, and it is anticipated that these compounds can be identified through isolation and purification based on the preliminary information presented.

### 2.5. Identification of Additional Flavonoid Glycosides

Quercetagetin-dimethyl is a relatively uncommon aglycone characterized by fragment ions at *m*/*z* 345, 330, and 315. Compounds **63**, **64**, and **65** were putatively characterized as quercetagetin-methyl derivatives, including *m*/*z* 345, 330, and 315 [40]. Compound **63** (RT 5.64 min) showed a [M−H]^−^ at *m*/*z* 507.114 and produced a fragment ion at *m*/*z* 345, indicating the loss of hexose (−162 Da). The glycosylation site was tentatively determined to be at the 3-OH position, based on the higher intensity of the [M−H−162]^•−^ ion. Consequently, compound **63** was putatively characterized as quercetagetin-dimethyl 3-*O*-hexoside by comparing its MS/MS spectrum with those reported in previous studies [40]. Compounds **64** and **65** were putatively characterized as quercetagetin-methyl derivatives based on their mass fragmentation patterns.

## 3. Discussion

Research on *Q. mongolica* pollen has been limited to data, particularly regarding its chemical composition. In this study, we performed untargeted chemical profiling using LC–MS/MS-based molecular networking to explore the metabolite diversity of *Q. mongolica* pollen, with a focus on flavonoid derivatives.

Flavonoids are well-known for their diverse biological activities. A previous study reported that 18 flavonoid metabolites isolated from *Q. mongolica* pollen exhibited strong antioxidant and tyrosinase inhibitory activities, supporting the functional potential of this material. In the present work, we tentatively annotated 69 flavonoid glycosides using GNPS-based molecular networking and spectral library matching. Based on their characteristic fragmentation patterns, kaempferol, quercetin, and isorhamnetin were identified as major aglycone backbones. *Q. mongolica* pollen was found to be particularly rich in isorhamnetin derivatives. The structural diversity of flavonoids observed highlights the importance of comprehensive metabolite profiling. Moreover, the presence of numerous previously unreported flavonoid glycosides suggests their potential as promising sources of novel bioactive compounds.

We anticipate that this study will lay the groundwork for future investigations into the bioactive metabolites of *Q. mongolica* pollen. Given that the chemical composition of *Q. mongolica* pollen can vary considerably depending on environmental factors and collection timing, the results of this study provide a crucial foundation for standardization studies. Standardization is essential to ensure consistent quality, reproducibility, and bioactivity in future applications, especially for pharmaceutical or functional food development. Thus, future research should aim to conduct standardization studies.

## 4. Materials and Methods

### 4.1. Reagents and Materials

HPLC-grade MeOH, H_2_O, and formic acid were obtained from Daejung Chemicals (Siheung, Republic of Korea). Two reference compounds, isorhamnetin 3-*O*-*β*-_D_-xylopyranosyl(1→6)-*β*-_D_-glucopyranoside and isorhamnetin 3-*O*-neohesperidoside, were isolated and structurally elucidated by 1D and 2D NMR spectroscopy (Appendix A).

### 4.2. Plant Materials and Extract Preparation

*Quercus mongolica* pollen collected from Yangyang, Gangwon-do, Republic of Korea, was purchased online (https://e-honey.net/goods/goods_view.php?goodsNo=127, accessed on 11 July 2023). The collected pollen was ground into a fine powder and 100 mg powder was extracted with 4 mL 80% MeOH at room temperature for 48 h. Following extraction, the solution was filtered and then concentrated under reduced pressure at 40 °C. An aliquot (5 mg) of the dried extract was dissolved in 5 mL of methanol and filtered through a 0.20 µm hydrophilic PTFE membrane filter (Advantec, Tokyo, Japan) prior to LC–MS/MS analysis.

### 4.3. UPLC–QTOF–MS/MS Analysis

The LC–MS/MS analysis was performed using an Agilent 1290 Infinity II UPLC system (Agilent Technologies, Santa Clara, CA, USA) coupled with a SCIEX ZenoTOF 7600 mass spectrometer (SCIEX, Framingham, MA, USA). Chromatographic separation was conducted on a Phenomenex Kinetex XB-C18 column (1.7 μm, 50 × 2.1 mm; Phenomenex, Torrance, CA, USA). The mobile phase comprised H_2_O (solvent A) and MeOH (solvent B) containing 0.1% formic acid. The procedure for gradient elution was as follows: 23–53% B (0–6 min), isocratic mode at 53% B (6–10 min), 53–100% (10–12 min), followed by washing with 100% B for 5 min and reconditioning with 23% B for 3 min. The flow rate was set to 0.3 mL/min, and the injection volume was 3 μL. UV detection was performed at 254 nm. MS data were collected in negative ionization mode using a TOF/MS–IDA–TOF MS/MS acquisition. The TOF/MS and TOF MS/MS ranges were set to 100−2000 and 80−2000 Da, respectively. The ion source parameters were as follows: curtain gas 35 psi; CAD gas 7; ion source gas 50 psi; source temperature 500 °C; spray voltage −4500 V. In the CID mode, 45 eV collision energy was applied. The chemical formula of each compound was predicted using the Formula Finder function in SCIEX OS software.

### 4.4. Data Processing

The raw LC–MS/MS data were acquired using SCIEX OS 3.3.1 software and subsequently converted to mzML files using the ProteoWizard MSconvert software version 3.0.24164. Data processing was carried out in MZmine 3.9.0, following the steps: peak detection, ADAP chromatogram builder, local minimum feature resolver, ^13^C isotope filter, and feature list row filter. The processed MS/MS spectra were exported as MGF files and uploaded to GNPS (http://gnps.ucsd.edu, accessed on 26 March 2025). A spectral similarity network was built using the GNPS. The GNPS parameters were as follows: precursor ion mass tolerance, 0.02 Da; MS/MS fragment ion tolerance, 0.02 Da. The edges were filtered to obtain a cosine score above 0.6 and a minimum of five matched peaks, and the spectral network was visualized using Cytoscape 3.10.2. The MS/MS molecular network was downloaded from https://gnps.ucsd.edu/ProteoSAFe/status.jsp?task=cf83418bc24746bcb152ee2d4d66b6dd (accessed on 26 March 2025).

## 5. Conclusions

This study provided the first untargeted metabolite profiling of *Q. mongolica* pollen using LC–MS/MS-based molecular networking. A total of 69 flavonoid glycosides were tentatively annotated, highlighting the chemical diversity and bioactive potential of this material. These findings lay the groundwork for future investigations into the bioactive metabolites of *Q. mongolica* pollen and standardization studies.

The flavonoid composition and abundance in bee pollen are known to vary significantly depending on environmental conditions, including geographical origin, cultivation practices, and the timing of pollen collection [7,8,27,28]. In this study, the pollen analyzed was collected from Yangyang, Gangwon-do, Republic of Korea, an area characterized by distinct climatic conditions that could influence secondary metabolite biosynthesis pathways, ultimately affecting flavonoid diversity and abundance. Future research incorporating samples collected from different regions or seasons would provide deeper insights into the environmental influences on flavonoid profiles and help guide standardization efforts.

## Figures and Tables

**Figure 1 ijms-26-07930-f001:**
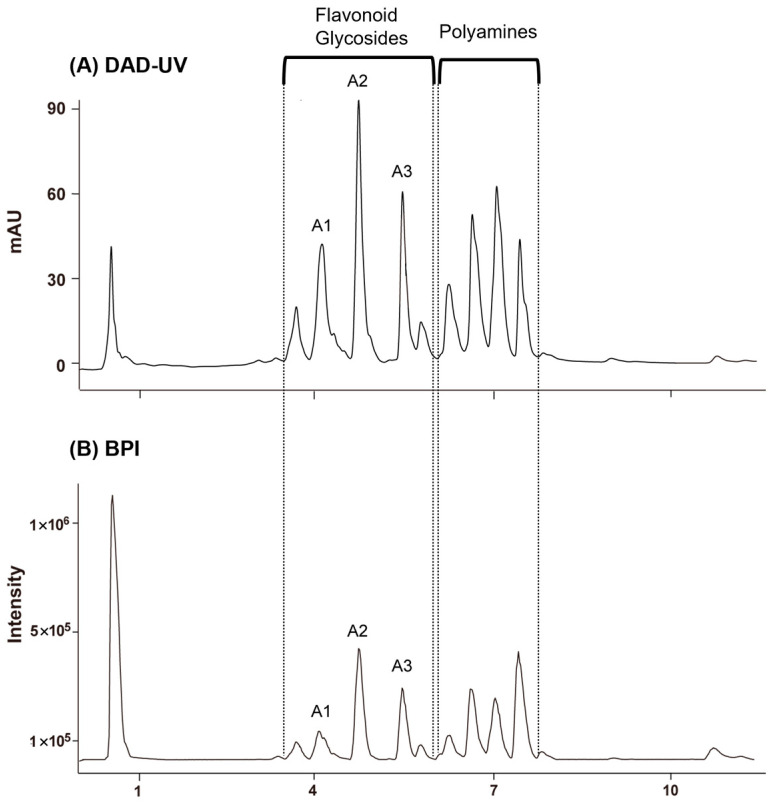
(**A**) DAD–UV chromatograms of *Quercus mongolica* pollen at 254 nm. (**B**) Base peak ion chromatogram (BPI) with reference compounds A1–A3. Peaks observed at 3.48–5.96 min were identified as flavonoid glycosides, while those at 6.01–7.68 min corresponded to polyamines.

**Figure 2 ijms-26-07930-f002:**
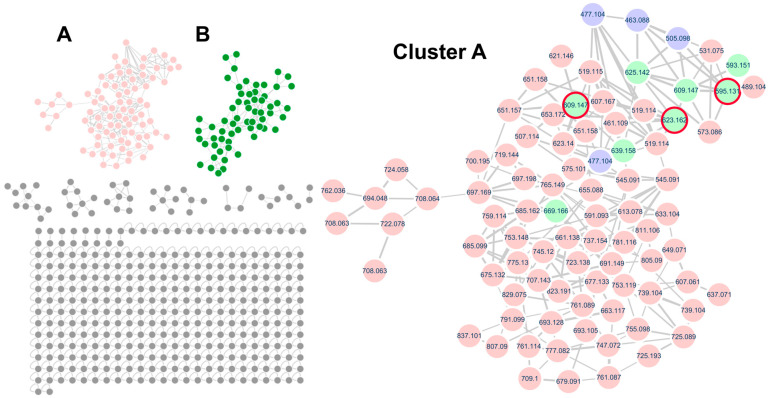
The MS/MS spectral molecular network of *Quercus mongolica* pollen in negative ion mode and the molecular family of flavonoid glycosides showing the molecular families of Cluster (**A**) (flavonoid glycosides, pink), Cluster (**B**) (polyamines, green). In Cluster (**A**), purple nodes, green nodes, and red circles represent flavonoid monoglycosides matched with the GNPS spectral library, flavonoid diglycosides, and reference compounds identified in *Q. mongolica* pollen, respectively.

**Figure 3 ijms-26-07930-f003:**
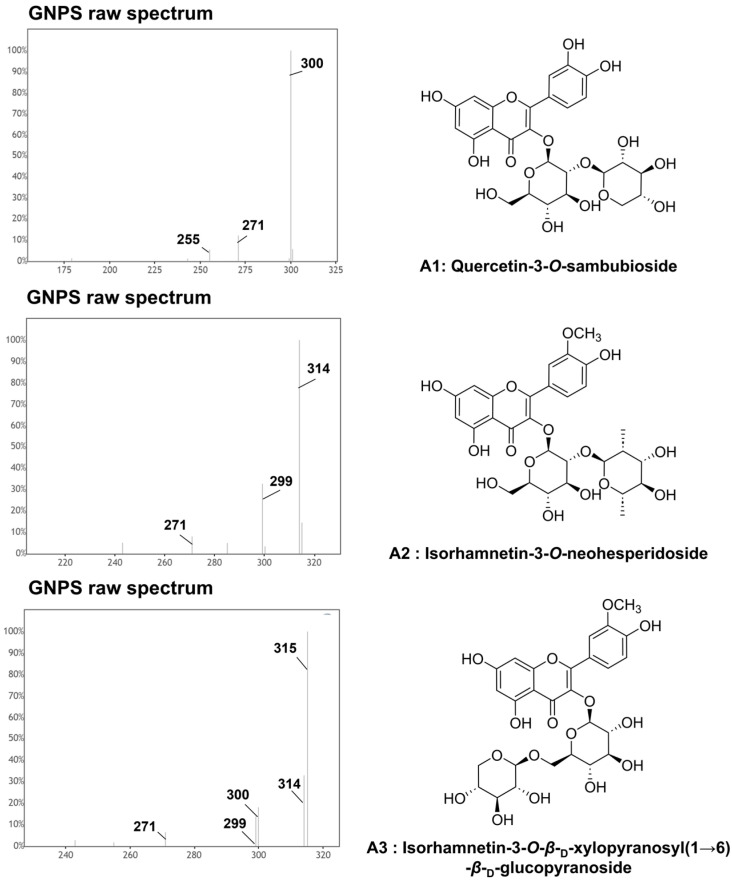
Raw MS/MS spectra of primary compounds (A1, A2, and A3) from *Quercus mongolica* pollen obtained from GNPS. Quercetin 3-*O*-sambubioside (A1, compound **6**; RT = 4.17 min; [M−H]^−^ *m*/*z* 595.132), isorhamnetin 3-*O*-neohesperidoside (A2, compound **30**; RT = 4.75 min; [M−H]^−^ *m*/*z* 623.165), isorhamnetin 3-*O*-*β*-_D_-xylopyranosyl(1→6)-*β*-_D_-glucopyranoside (A3, compound **28**; RT = 5.47 min; [M−H]^−^ *m*/*z* 609.165).

## Data Availability

Raw NMR data for natural products were deposited in the Harvard Dataverse (https://dataverse.harvard.edu, accessed on 12 August 2025) and are accessible at DOI: [https://doi.org/10.7910/DVN/MYK0KV].

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
