# Peer review of "Molecular Networking-Guided Annotation of Flavonoid Glycosides from Quercus mongolica Bee Pollen"

_ijms, 2025, doi:10.3390/ijms26167930_

Round 1
Reviewer 1 Report
Comments and Suggestions for Authors
The article titled “Molecular networking-guided annotation of flavonoid glycosides from Quercus mongolica bee pollen” presents a thorough study focused on the annotation of flavonoid glycosides in Quercus mongolica bee pollen using LC-MS/MS-based molecular networking. The integration of GNPS spectral library searches with characteristic MS/MS fragment interpretation and comparison to reference standards is a notable strength and exemplifies best practices in untargeted metabolomics. The authors correctly emphasize the value of glycosylation site determination and provide clear rationale for aglycone backbone identification via spectral fragmentation trends. While the study benefits from a commendable level of detail in compound annotation, including multiple isomeric structures and uncommon flavonoid variants, the manuscript occasionally suffers from redundancy, particularly in the descriptions of glycosylation positions and MS/MS interpretation trends, which could be condensed for better readability. The work would benefit from a stronger contextualization of the flavonoid glycoside diversity in terms of ecological function or potential pharmacological implications. Additionally, while the findings are solid, the significance of flavonoid diversity could be further underscored by comparative profiling across related species or multiple sample sources to assess the consistency and novelty of the reported compounds. The manuscript also lacks a more in-depth critical evaluation of the limitations of molecular networking (e.g., challenges in resolving isomers or dependence on spectral library coverage), although this is briefly acknowledged in the final compound annotations. Figures are generally clear and appropriate, though some would benefit from improved resolution and more detailed labeling for easier interpretation. Tables and compound attributions are helpful, though future studies may further substantiate annotations with MS/MS-based structure elucidation software or computational metabolite prediction tools. Importantly, the study identifies multiple previously unreported glycosides, which provides a compelling case for continued research into bee pollen metabolomics. Overall, the manuscript represents a valuable addition to the field of natural product chemistry and plant metabolomics, offering foundational insights into the specialized metabolites of Q. mongolica pollen. The reviewer has the following comments that authors need to address:
- In Figure 1, clearer peak annotations for compounds A1–A3 would enhance interpretability. Additionally, an explanation for the additional peak observed around 14.5 minutes in the BPI chromatogram is needed to clarify.
- Justify why only two reference compounds were confirmed by NMR and clarify whether selective validation of additional compounds was considered feasible. Also, in the 1H NMR spectra, the authors should provide the integration values and the interpretation of COSY, HSQC and HMBC data should be presented. This would strengthen the reliability and interpretative depth of the analytical approach.
Author Response
Reviewer 1
Comments
The article titled “Molecular networking-guided annotation of flavonoid glycosides from Quercus mongolica bee pollen” presents a thorough study focused on the annotation of flavonoid glycosides in Quercus mongolica bee pollen using LC-MS/MS-based molecular networking. The integration of GNPS spectral library searches with characteristic MS/MS fragment interpretation and comparison to reference standards is a notable strength and exemplifies best practices in untargeted metabolomics. The authors correctly emphasize the value of glycosylation site determination and provide clear rationale for aglycone backbone identification via spectral fragmentation trends. While the study benefits from a commendable level of detail in compound annotation, including multiple isomeric structures and uncommon flavonoid variants, the manuscript occasionally suffers from redundancy, particularly in the descriptions of glycosylation positions and MS/MS interpretation trends, which could be condensed for better readability. The work would benefit from a stronger contextualization of the flavonoid glycoside diversity in terms of ecological function or potential pharmacological implications. Additionally, while the findings are solid, the significance of flavonoid diversity could be further underscored by comparative profiling across related species or multiple sample sources to assess the consistency and novelty of the reported compounds. The manuscript also lacks a more in-depth critical evaluation of the limitations of molecular networking (e.g., challenges in resolving isomers or dependence on spectral library coverage), although this is briefly acknowledged in the final compound annotations. Figures are generally clear and appropriate, though some would benefit from improved resolution and more detailed labeling for easier interpretation. Tables and compound attributions are helpful, though future studies may further substantiate annotations with MS/MS-based structure elucidation software or computational metabolite prediction tools. Importantly, the study identifies multiple previously unreported glycosides, which provides a compelling case for continued research into bee pollen metabolomics. Overall, the manuscript represents a valuable addition to the field of natural product chemistry and plant metabolomics, offering foundational insights into the specialized metabolites of Q. mongolica pollen. The reviewer has the following comments that authors need to address:
Reply/Action
Thank you very much for clearly explaining the significance and content structure of this research, as well as the aspects that need to be revised going forward. Regarding your comments, the results have been explained with essential MS/MS fragmentation values for each fraction (quercetin, kaempferol, isorhamnetin, and other flavonoid aglycon parts) first, and the glycosylation part next. This sorting will help distinguish the glycosylation types based on the aglycon structure. Also, Figure 1 has added annotations for compounds, classes of structure, and the removal of unnecessary parts (after 12min) to clarify the interpretation of the chromatogram.
- In Figure 1, clearer peak annotations for compounds A1–A3 would enhance interpretability. Additionally, an explanation for the additional peak observed around 14.5 minutes in the BPI chromatogram is needed to clarify.
Reply/Action #1
- In Figure 1, annotations for compounds A1–A3, class of the structure have been indicated. Also, the following sentence has been added in Page 3, Line 128: “Figure 1 shows the base peak ion chromatogram (BPI) of Q. mongolica According to LC–MS/MS analysis, peaks eluting between 3.5 and 6.0 min were identified as flavonoid glycosides, whereas those between 6.0 and 7.8 min were attributed to polyamine compounds. Three major flavonoid glycosides were shown the base peak ion chromatogram (BPI) of Q. mongolica pollen, in which three primary flavonoid glycosides were prominently observed (A1–A3).”
- Peaks detected at approximately 13.0–14.0 min correspond to the washing step with 100% MeOH (12–17 min). These types of compounds are more nonpolar. The peaks at 13.04 min and 13.13 min, observed at m/z 277.217 [M−H]− (RT= 13.04 min, calculated 278.436, C₁₈H₃₀O₂) and 279.234 [M−H]− (RT= 13.13 min, calculated 280.452, C₁₈H₃₄O₂) in LC-MS, indicate fatty acid derivatives, specifically linolenic acid and linoleic acid. Regarding the UV-DAD, which indicated weak absorption, it was anticipated that those peaks could also be fatty acids.
- Figure 1 has been edited to improve the interpretation of the chromatogram by removing unnecessary sections, specifically after 12 minutes of the chromatogram, and evidence of the interpretation of the nonpolar region has been added on Page 4.
Retention time = 13.04 min; [M−H]− m/z 277.217; linolenic acid)
Retention time = 13.13 min; [M−H]− m/z 279.234; linoleic acid)
- Justify why only two reference compounds were confirmed by NMR and clarify whether selective validation of additional compounds was considered feasible. Also, in the 1H NMR spectra, the authors should provide the integration values and the interpretation of COSY, HSQC and HMBC data should be presented. This would strengthen the reliability and interpretative depth of the analytical approach.
Reply/Action #2
- Chemical assignment of 1H and 13C NMR spectral data for two reference compounds has been added to Table S1. Also, integration values and interpretation of COSY, HSQC and HMBC spectra have been edited into the supporting information. (Table S1, Figure S3−S7 for compound 28 and S9−S13 for compound 30)
- The selection of two reference compounds has been clearly isolated from other flavonoid glycoside derivatives based on UPLC-PDA and LC-MS analysis. Identification of flavonoid glycosides is not affected by overlaps in the chromatogram. However, for utilizing bee pollen resources, it is important to standardize using the marker compounds. As mentioned, compounds A2 and A3 show potential to be used as marker compounds in future uses, as they are well isolated from the other compounds. Also, a few sentences have been edited on Page 4, Line 154: “The chemical composition of bee pollen varies depending on geographic origin, collection period, and cultivation conditions, factors which are also known to influence its biological activities. Therefore, standardization studies are necessary. Two primary compounds, isorhamnetin 3-O-β-D-xylopyranosyl (1→6)-O-β-D-glucopyranoside and isorhamnetin 3-O-neohesperidoside, can be used as standard compounds for future standardization research. These compounds were selected because they are among the major constituents of mongolica pollen, and their chromatographic signals were well separated from other peaks, making them suitable for accurate quantitative analysis.”

Reviewer 2 Report
Comments and Suggestions for Authors
Using Quercus mongolica bee pollen as the study material, the authors identified 69 flavonoid glycosides—2 kaempferol, 14 quercetin, and 46 isorhamnetin derivatives—by UPLC-QTOF-MS/MS coupled with GNPS molecular networking, and confirmed the structures of the two main compounds with authentic standards. The dataset is comprehensive and offers a fresh perspective for the chemical characterization of bee pollen. During manuscript preparation the authors should address the following points:
(1) The term “molecular networking” is not emphasized in the Results; we recommend revising the title to “Identification of Flavonoids in Quercus mongolica Bee Pollen.”
(2) In the Keywords section of the Abstract, “pollen” should be listed as an independent keyword.
(3) In Figure 1, please label each peak with its corresponding number and annotate the compound it represents.
(4) References should follow a uniform style: add DOIs (e.g., refs. 1, 2, 5); include issue numbers and article IDs or page ranges (e.g., refs. 1, 3); and italicize Latin binomials (e.g., refs. 1, 15).
Author Response
Reviewer 2
Comments
- The term “molecular networking” is not emphasized in the Results; we recommend revising the title to “Identification of Flavonoids in Quercus mongolica Bee Pollen.”
Reply/Action #1
Thank you for pointing out the lack of information in the results. Instead of replacing the title, I decided to add more information about GNPS on the 3.1 Flavonoid glycoside profiling of Q. mongolica pollen using molecular networking based on UPLC–QTOF–MS/MS analysis. I added a few more sentences to the manuscript and emphasized the benefits of GNPS to explore the undiscovered metabolites, which are limited by the lack of quantity, stability, and overlap with the isolation techniques.
- The Keywords section of the Abstract, “pollen” should be listed as an independent keyword.
Reply/Action #2
We appreciate this suggestion. The keywords have been separated into “Quercus mongolica” and “Pollen” as requested.
- In Figure 1, please label each peak with its corresponding number and annotate the compound it represents.
Reply/Action #3
We appreciate this suggestion. In Figure 1, annotations for compounds A1–A3 and the class of structure have been indicated.
- References should follow a uniform style: add DOIs (e.g., refs. 1, 2, 5); include issue numbers and article IDs or page ranges (e.g., refs. 1, 3); and italicize Latin binomials (e.g., refs. 1, 15).
Reply/Action #4
We appreciate this suggestion. All the references have been updated with uniform DOIs, including issue numbers and article IDs or page ranges, and Latin binomials are italicized.

Reviewer 3 Report
Comments and Suggestions for Authors
The submitted manuscript entitled “Molecular Networking–Guided Annotation of Flavonoid Glycosides from Quercus mongolica Bee Pollen,” by Yerim Joo, Eunbeen Shin, Hyunwoo Kim, Mi Kyeong Lee, and Seon Beom Kim, makes a significant contribution to the identification of flavonoid glycosides from natural products. The manuscript is clearly and concisely written, the references are appropriate, and the authors have a respectable publication record in this area. In view of these points, I consider the manuscript to be of interest to readers and researchers, as it provides new insights into the identification of flavonoid glycosides.
I have one main suggestion. In the Introduction, please add a few sentences on the significance of these compounds, which would broaden readers’ interest; the authors themselves have relevant publications on the biological activities of these compounds that could be briefly cited. Furthermore, as the authors note, the flavonoid profile and abundance depend on numerous factors; it would be helpful to include a short discussion of how these factors may pertain to the sample investigated.
Author Response
Reviewer 3
Comments
- I have one main suggestion. In the Introduction, please add a few sentences on the significance of these compounds, which would broaden readers’ interest; the authors themselves have relevant publications on the biological activities of these compounds that could be briefly cited.
Reply/Action #1
We appreciate this suggestion. In the Introduction, we have added additional context emphasizing the biological significance of flavonoids, highlighting their antioxidant, anti-inflammatory, antibacterial, and anticancer effects, as reported in previous studies, including our earlier publications (doi: 10.3390/molecules30040794), (Page 2, Line 44). Specifically, we cited our prior work demonstrating strong antioxidant and tyrosinase-inhibitory activities of flavonoids isolated from Quercus mongolica pollen, reinforcing their potential applications in pharmaceuticals, nutraceuticals, and functional foods. These additions provide a broader context and underscore the importance of understanding flavonoid composition and diversity in bee pollen.
- Furthermore, as the authors note, the flavonoid profile and abundance depend on numerous factors; it would be helpful to include a short discussion of how these factors may pertain to the sample investigated.
Reply/Action #2
We appreciate this valuable suggestion. We have expanded the Discussion section (Page 11, Line 329) to briefly highlight how environmental factors such as geographical origin, collection period, and cultivation conditions influence flavonoid composition, relating specifically to the pollen sample investigated. In this study, the pollen was collected from Yangyang, Gangwon-do, South Korea, a region known for its specific climate conditions, including temperature variation and humidity, which likely influence secondary metabolite production in pollen. Recognizing these factors is critical for standardizing quality control and understanding seasonal or geographical variability in pollen-derived bioactive compounds.
